# Subcutaneous Stromal Cells and Visceral Adipocyte Size Are Determinants of Metabolic Flexibility in Obesity and in Response to Weight Loss Surgery

**DOI:** 10.3390/cells11223540

**Published:** 2022-11-09

**Authors:** Séverine Ledoux, Nathalie Boulet, Chloé Belles, Alexia Zakaroff-Girard, Arnaud Bernard, Albéric Germain, Pauline Decaunes, Anaïs Briot, Jean Galitzky, Anne Bouloumié

**Affiliations:** 1Explorations Fonctionnelles, Hôpital Louis Mourier (APHP.Nord) and University Paris Cité, 92700 Colombe, France; 2Institute of Metabolic and Cardiovascular Diseases, INSERM, UMR1297, Team Dinamix, 31432 Toulouse, France; 3UMR 1231 Lipides/Nutrition/Cancer INSERM/Univ Bourgogne-Franche-Comté/AgroSupDijon, 21000 Dijon, France

**Keywords:** hypertrophy, hyperplasia, adipogenesis, lymphocytes, fibrosis, macrophages

## Abstract

Adipose tissue (AT) expansion either through hypertrophy or hyperplasia is determinant in the link between obesity and metabolic alteration. The present study aims to profile the unhealthy subcutaneous and visceral AT (SAT, VAT) expansion in obesity and in the outcomes of bariatric surgery (BS). The repartition of adipocytes according to diameter and the numbers of progenitor subtypes and immune cells of SAT and VAT from 161 obese patients were determined by cell imaging and flow cytometry, respectively. Associations with insulin resistance (IR) prior to BS as well as with the loss of excessive weight (EWL) and IR at 1 and 3 years post-BS were studied; prior to BS, SAT and VAT, unhealthy expansions are characterized by the accumulation of adipogenic progenitors and CD4+ T lymphocytes and by adipocyte hypertrophy and elevated macrophage numbers, respectively. Such SAT stromal profile and VAT adipocyte hypertrophy are associated with adverse BS outcomes. Finally, myofibrogenic progenitors are a common determinant of weight and IR trajectories post-BS; the study suggests that adipogenesis in SAT and adipocyte hypertrophy in VAT are common determinants of metabolic alterations with obesity and of the weight loss and metabolic response to bariatric surgery. The data open up new avenues to better understand and predict individual outcomes in response to changes in energy balance.

## 1. Introduction

Obesity is the unique symptom of a myriad of subtle phenotypes with different weight trajectories, fat repartitions and risks of developing metabolic disturbances. Central obesity, associated with excess visceral fat, has adverse consequences for cardio-metabolic health [1]. Impaired capacity of the subcutaneous adipose tissue (SAT) to buffer lipids in excess together with increased free fatty acid turnover from visceral adipose tissue (VAT) depots are thought to be part of the mechanisms responsible for the metabolic vulnerability leading to insulin resistance (IR) and type 2 diabetes [2]. Resident progenitors are key cellular components in the maintenance of tissue integrity and function. Single-cell RNA sequencing [3] as well as flow cytometry approaches [4] clearly highlight distinct progenitor subsets triggering and/or regulating adipogenic fate, with fat depot-specific origin, niche and repartition. Impaired adipogenesis precludes adipocyte renewal and hyperplasia [5]. Consequently, the accumulation of dysfunctional hypertrophied adipocytes promotes an unresolved low-grade inflammatory state with the accumulation of immuno-inflammatory cells further contributing to the local imbalance between adipogenesis and myofibrogenesis [6]. Therefore, the phenotypes of both cell compartments composing fat depot, i.e., the adipocyte and the stromal cells including progenitor and immune cells, are key components defining the healthy AT expansion associated with metabolic flexibility [5,7]. As the expansion of SAT or VAT are not associated with the same risks for metabolic health [8], it is important to better define the fat depot-specific phenotypes and to establish whether they are also critical under negative energy balance as is promoted by weight loss surgery. Bariatric surgery (BS) represented mainly by Roux-en-Y gastric bypass (RYGB) and sleeve gastrectomy (SG), is currently the most efficient long-term treatment of severe obesity. The extent of weight loss (WL) reached after BS remains highly variable [9]. About 30% of patients with insufficient weight loss (defined by an excess weight loss (EWL) less than 50%) will need additional procedure [10], whereas up to 2/3 regain weight in the long term [11]. In the same line, despite a rapid improvement of metabolic parameters after surgery, persistent IR is frequently observed, with less than half of the subjects achieving diabetes remission in the long term [12]. The predictive factors of EWL and IR improvement after BS are still not well defined [10,13]. Age, high preoperative body mass index (BMI), type 2 diabetes and visceral adiposity are all reported to negatively impact postoperative WL [14]. Few data are available concerning the impact of matched SAT and VAT expansion phenotypes on BS success, especially in the long term [15], but it is assumed that patients with unhealthy SAT and/or VAT expansion will have not only more baseline metabolic disturbances but also less chance to correct obesity and diabetes after BS. Conflicting results have been reported on adipocyte hypertrophy [16], AT inflammation [13,17] or frequency of progenitor cells in the stroma [18] on WL or diabetes remission at 1 year after BS, and as far as we know, no data are available concerning the numbers of the immune cell subtypes in human AT and the distinct progenitor subsets. 

In the present study, we explore the repartition of adipocyte diameters and the stroma progenitor subsets and immune cells using flow cytometry approaches in matched SAT and VAT from a large cohort of patients with severe obesity. We studied the relationships between SAT and VAT cell phenotypes with IR and diabetes prior surgery and with weight loss and metabolic outcomes 1 year and 3 years post-surgery. Our data highlight depot-specific cell features defining unhealthy fat mass expansion associated with impaired metabolic flexibility that are common with obesity and in response to weight loss surgery. 

## 2. Materials and Methods

### 2.1. Human Cohorts and Tissue Collection

All clinical and biological data were obtained from the SENADIP cohort (ClinicalTrials.gov Identifier: NCT01525472) including consecutive patients with severe obesity (BMI > 40 kg/m*^2^* or BMI > 35 kg/m*^2^* with comorbidities) who underwent BS (Roux-en-Y gastric bypass or sleeve gastrectomy) in the same hospital, between April 2012 and October 2015, after a nutritional preparation of at least 6 months, in accordance with recommendations and international guidelines of the French health authority (HAS) [19,20]. The surgical techniques are described elsewhere [20]; SAT- and VAT-matched biopsies were obtained during surgery as previously described [4]. Clinical (anthropometric parameters, comorbidities, medications) and routine biological parameters (including 2 measures of fasting blood glucose and insulin for calculation of HOMA-IR = (glucose (mmol/l) × insulin (mU/l))/22.5, with HOMA-IR > 2.4 defining insulin resistance (IR)) were collected, as previously described [20], in the month before surgery and then systematically at 1 year and 3 years after surgery during a one-day hospitalization in the same medical center. The ethical committee approved the protocol, and all donors gave their written informed consent before surgery. The baseline characteristics of the 161 patients included in the present study are detailed in Table 1, with 47% of the individuals with metabolic syndrome as defined by NCEP-ATP III 2005, with 17% treated for type 2 diabetes, 25% for hypertension, 14% for dyslipidemia and 31% with sleep apnea obstructive syndrome. One hundred and fifty-three patients were present for the visit scheduled 1 year (12 ± 3 months) after surgery and 78 patients for the visit at 3 years (36 ± 2 months). The main surgical procedure was Roux-en-Y gastric bypass (RYGB) (71% of the patients). Outcomes of BS on weight loss and metabolic parameters are indicated in Table 1.

### 2.2. Adipose Tissue Digestion and Isolation of Adipocytes and Stromal Cells

Tissue biopsies were sequentially digested with type I collagenase (250 U/mL, Sigma-Aldrich). After digestion and centrifugation, the upper phase containing mature adipocytes was retrieved and the erythrocyte lysis step was performed on a pellet containing stroma–vascular cells followed by serial filtrations. The viable recovered cells were counted and further analyzed by flow cytometry. Photographs of mature adipocytes were taken by phase microscopy and cell diameter determined by image analysis using NIKON NIS software.

### 2.3. Flow Cytometry Analyses

One hundred thousand cells were incubated with three distinct panels with fluorescent-labeled antibodies: panel 1 for progenitor cell subtypes PE-MSCA1, PeVio770-CD14, APC-CD271 (Miltenyi Biotec, Paris, France); PerCP-CD34, V450-CD31, BV510-CD45 (BD Biosciences, Le Pont De Claix, France); panel 2 for macrophage subtypes: PeVio770-CD14, APC-CD206, BV510-CD45 (BD Biosciences); panel 3 for lymphocytes FITC-CD4, PerCP-CD8, Pe-Cy7-CD56; APC-Cy7-CD19, V450-CD3; BV510-CD45 (BD Biosciences, Le Pont De Claix, France) or an isotype control panel for 30 min at 4 °C in phosphate buffered saline (PBS, Eurobio Scientific, Les Ulis, France) supplemented with 0.5% bovine serum albumin (BSA, Gibco, ThermoFisher Scientific, Les Ulis, France) and 2 mmol/L ethylenediaminetetraacetic acid (EDTA). Cells were washed with PBS and analyzed using a FACS Canto^TM^ II flow cytometer and Diva Pro software (BD Biosciences, Le Pont De Claix, France). Cell subsets were identified through the combination of cell surface markers as described in Table 2 and the gating strategy shown in Figure 1B.

### 2.4. Statistical Analyses

Statistical analyses were performed using Prism (Prism 9, GraphPad Software, San Diego, USA). Comparisons between two groups were analyzed either by two-tailed paired or unpaired Student’s t-test. Comparisons between more than two groups were analyzed by one-way or two-way ANOVA followed by appropriate post-tests for (*n*) independent experiments. Correlation matrix was plotted by the R package corrplot (R studio) using Spearman. Differences are considered statistically significant when *p* < 0.05. Multiple linear regression analyses were performed for studying the relation between WL and HOMA-IR index at 3 years and baseline AT characteristics. AT components that were significantly associated (*p* < 0.05) in univariate analysis were included in the model and adjusted to baseline clinical characteristics (including age, BMI, waist-hip ratio, HOMA-IR) and WL at 3 years for HOMA-IR at 3 years. Only patients with available data were included in the analysis at 1 and 3 years.

## 3. Results

### 3.1. Fat Depot-Specific Cell Phenotypes in Patients with Severe Obesity

The repartition according to diameter of mature adipocytes of matched biopsies of SAT and VAT from obese patients was shifted towards hypertrophied (from 120 to 150 µm diameter) and small adipocytes (20 and 30 µm of diameter) in SAT compared to VAT (Figure 1A). VAT adipocytes exhibited a higher proportion of cells with intermediate diameters (from 70 to 100 µm) with a median of 89.2 ± 13.0 vs. 93.9 ± 15.8 µm for VAT and SAT, respectively (*p* = 0.0059) (Figure 1A).

In parallel, the stroma phenotype analyzed by flow cytometry showed marked fat depot specificity (Figure 1B). The repartition of the subsets of progenitors was in favor of both adipogenic and immature progenitors in SAT while in favor of the myofibrogenic subset in VAT. Concerning the immune landscape, VAT is characterized by higher numbers of innate lymphocytes including NK and NKT cells and adaptive helper and cytotoxic T and B lymphocytes. Of note, no difference in fat depot was observed for macrophage numbers, neither resident nor recruited (Figure 1B).

### 3.2. Cell Phenotypes of Expanded SAT and VAT Associated with Impaired Metabolic Flexibility in Obesity

The associations between the parameters of insulin resistance (HOMA-IR index and HbA1c) and SAT/VAT adipocyte diameters and stromal cell numbers are detailed in Figure 2A,B, respectively. Although discrete associations were detected either with HOMA-IR or with HbA1c, the correlations were considered relevant for impaired metabolic flexibility when present for both parameters. Concerning the repartition of the adipocytes according to diameter, both HOMA-IR and HbA1c parameters were positively associated with VAT hypertrophied adipocytes (with diameters from 120 µm to more than 130 µm) and negatively with VAT adipocytes of intermediate diameters (from 70 µm to 90 µm). High numbers of SAT adipogenic progenitors and T lymphocytes (either helper or cytotoxic) were positively associated with both IR parameters while elevated VAT resident and recruited macrophage numbers are associated with HOMA-IR and HbA1c, respectively. The comparison of both SAT and VAT cell phenotypes between diabetic and non-diabetic patients, whose clinical characteristics are indicated in Table 3, highlights a marked shift in the repartition of the VAT adipocytes towards hypertrophic adipocytes (from 110 µm to 130 µm) with a concomitant reduction in the percentage of adipocytes with intermediate diameters (70 µm to 100µm) for diabetic patients.

No differences were found for the SAT adipocytes (Figure 2C). The SAT of diabetic patients contained higher numbers of adipogenic progenitors and helper T lymphocytes while their VAT was characterized by more recruited macrophages and B lymphocytes (Figure 2D).

**Figure 2 cells-11-03540-f002:**
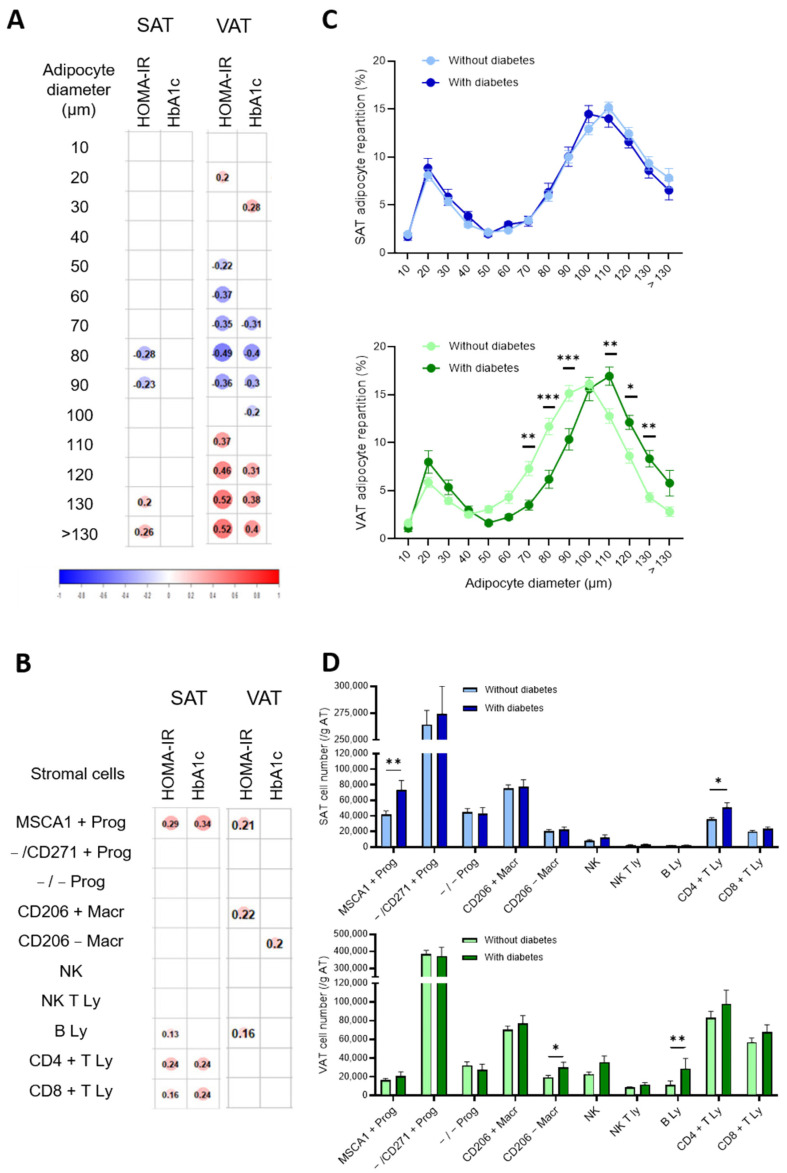
Cell phenotypes of expanded SAT and VAT associated with impaired metabolic flexibility prior to bariatric surgery. Correlation matrix between mature adipocyte diameter repartition (%) in SAT and VAT (**A**) or stromal cell numbers determined by flow cytometry analyses (**B**) and HOMA-IR and HbA1c (%). Spearman correlations with *p* < 0.05 are shown with Spearman r value indicated in the color circle for *n* = 101 for adipocyte diameters and *n* = 135 to 140 patients for stromal cells. (**C**) Adipocyte diameter repartition in SAT (blue) and VAT (green) of patients with (dark) or without (light) diabetes. Values are means ± SEM of *n* = 74 without diabetes and *n* = 28 with diabetes, two-way ANOVA followed by Sidak multiple comparisons test, *, *p* < 0.05, **, *p* < 0.01, ***, *p* < 0.001. (**D**) Stromal cell numbers determined by flow cytometry analyses in SAT (blue) and VAT (green) of patients with (dark) or without (light) diabetes. Values are means ± SEM of *n* = 101 to 105 without diabetes and *n* = 34 to 35 with diabetes, multiple Mann-Whitney tests, *, *p* < 0.05, **, *p* < 0.01.

When the results obtained with correlation analysis and the ones with the comparison between groups are taken together, common cell phenotypes emerge as characteristics of unhealthy AT expansion: accumulation of adipogenic progenitors and helper T lymphocytes in SAT and elevated adipocyte hypertrophy and accumulation of recruited macrophages in VAT.

### 3.3. Cell Phenotypes of Expanded SAT and VAT Associated with Weight Loss at 1 and 3 Years after Bariatric Surgery

Correlation matrix of the SAT and VAT cell phenotype*s* with EWL at one and three years post-BS are presented in Figure 3.

The percentages of VAT hypertrophied adipocytes (from 130 µm to more than 130 µm) were negatively correlated with EWL at one and three years post-BS while the ones of adipocytes with intermediate diameters (80 µm to 100 µm) were positively associated (Figure 3A). Positive but weak correlations were also observed for SAT adipocytes of intermediate diameter (100 µm) with EWL at one and three years (Figure 3A). In multiple linear regression analysis, after adjustment for baseline characteristics (including BMI, age and HOMA-IR), EWL at three years was still correlated with the proportion of large adipocytes in VAT only (*p* = 0.026). Concerning the number of stromal cells, the sole population showing association with EWL whatever the time period post-surgery was the SAT adipogenic progenitor (Figure 3B). SAT higher numbers of B and T (CD4 +, CD8 + and NKT) lymphocytes were associated with lower weight loss after three years only while VAT immature progenitor numbers showed a weak positive association during the same time period (Figure 3B). Patients were then grouped according EWL failure or success, with EWL failure defined as EWL less than 50%. Sixteen percent of the patients failed to lose excess weight at three years. They had higher HbA1c prior to surgery but no differences in age, BMI or HOMA-IR compared with patients with WL success (Table 4).

As observed in the correlation matrix, the comparison between the groups of patients highlights differences in adipocyte repartition according to diameter only in VAT (Figure 3C), with a shift towards hypertrophied adipocytes in patients with EWL failure, while the patients with EWL success exhibited higher percentages of VAT adipocytes with intermediate diameter (80 µm and 90 µm, *p* < 0.05 and 0.01, respectively) (Figure 3C). Similarly, in agreement with the correlation matrix, the patients with EWL failure had a higher number of SAT adipogenic progenitors as well as B and helper and cytotoxic T lymphocytes (Figure 3D). No differences in VAT cell numbers were detected (Figure 3D). Therefore, the accumulation of adipogenic progenitors as well as those of B, helper and cytotoxic T lymphocytes in expanded SAT and elevated adipocyte hypertrophy in expanded VAT prior to surgery are predictors of failure to lose excess weight in response to bariatric surgery.

### 3.4. Weight Trajectories between 1 and 3 Years Post-Bariatric Surgery

To investigate potential impacts of expanded SAT and VAT cell phenotypes on the weight trajectories between 1 and 3 years post-surgery, patients were stratified in tertiles based on the difference between EWL at 3 years and EWL at 1 year after BS (Figure 4), with the first tertile being patients with continuous weight lost, the second being patients who stabilized their weight between the two time points and the third being patients who regained weight at three years compared to one year (Figure 4A).

At one year post-BS, EWL was not different whatever the groups. At three years, the highest EWL was found in patients with weight loss trajectory (*p* < 0.0001 compared to stable weight and weight regain trajectories). The repartitions in adipocyte diameters in both SAT and VAT were not different whatever the trajectories (Figure 4C) but both SAT and VAT numbers of myofibroblastic progenitors were lower in the weight loss trajectory compared to the stable weight and weight regain trajectories (Figure 4B).

### 3.5. Cell Phenotypes of Expanded SAT and VAT Associated with Failure to Improve Insulin Resistance at 1 and 3 Years Post-Bariatric Surgery

HOMA-IR and HbA1c significantly decreased (*p* < 0.001) at 1 and 3 years (Table 1) post-surgery. Correlation matrix analysis of the SAT and VAT cell phenotypes with HOMA-IR and HbA1c at 1 and 3 years is presented in Figure 5.

Although discrete associations were detected either with HOMA-IR or with HbA1c, the correlations were considered as relevant for improved metabolic flexibility when present for both parameters whatever the time period post-BS. The repartition in adipocyte diameters exhibited the strongest association with HOMA–IR and HbA1c at 1 year and three years post-BS in VAT only. Indeed, percentages of VAT-hypertrophied adipocytes (130 µm and more than 130 µm) are associated with impaired metabolic flexibility while the ones of VAT intermediate adipocytes (70/80µm to 90/100 µm) correlate with improved metabolic flexibility (Figure 5A). High numbers of SAT adipogenic progenitors are associated with impaired metabolic flexibility whatever the time period post-BS while the ones of SAT NKT and helper T lymphocytes are associated at three years post-BS (Figure 5B). After adjustment for baseline characteristics (age, BMI and HOMA-IR) and for weight loss at 3 years, HOMA-IR at 3 years remained predicted by the number of adipogenic progenitor cells in SAT (*p* = 0.010) but not by the number of SAT lymphocytes subsets.

Three years post-surgery, in the 68 subjects of 78 from whom the HOMA index was available, 28% of the patients exhibited persistent IR. They had higher pre-operative HOMA-IR as well as HbA1c and lost less weight post-BS (Table 5).

In agreement with the correlation matrix, a shift towards hypertrophied adipocytes was observed in VAT of patients with persistent IR (130 µm to more than 130 µm) (Figure 5C) while patients with improved metabolic flexibility had greater percentages of VAT adipocytes with intermediate diameter (80 µm and 90 µm, *p* < 0.05 and 0.01, respectively) (Figure 5C). Similarly, as was observed in the correlation matrix, the patients with persistent IR had a higher number of SAT adipogenic MSCA1 + progenitors as well as NKT and CD4 + T lymphocytes (Figure 5D).

### 3.6. Insulin Resistance Trajectories between 1 and 3 Years Post-Bariatric Surgery

To investigate potential impacts of expanded SAT and VAT cell phenotypes on the metabolic trajectories between 1 and 3 years post-surgery, patients were stratified in tertiles based on the difference of HOMA-IR at 3 years and HOMA-IR at 1 year after BS, with the first tertile being patients who further improved IR, the second being patients with stable HOMA-IR and the third being patients whose IR worsened (Figure 6A). The HOMA-IR index prior to surgery was not different between groups and all groups exhibited an improved HOMA-IR index at 1 and 3 years after surgery (*p* < 0.0001). HOMA-IR was lower at 3 years after BS in both IR improvement and stable trajectories compared to the IR deterioration trajectory (*p* = 0.0022 and *p* = 0.0003, respectively), despite a higher mean value at 1 year after surgery in the group with favorable trajectory (*p* = 0.0002 compared to stable, *p* = 0.002 compared to negative). The repartitions in adipocyte diameter in the SAT and VAT were not different whatever the trajectories (Figure 6C). Patients with a favorable IR trajectory were characterized by lower numbers of SAT myofibroblastic progenitors and resident macrophages (Figure 6B).

## 4. Discussion

The present study demonstrates that the adipocyte size and the numbers of progenitor subsets and immune cells are determinant not only in the metabolic alterations associated with AT expansion in obesity, but also in the outcomes of weight loss surgery in terms of EWL and improvement of IR. The data clearly highlight fat depot-specific cell compartments, i.e., adipocytes in VAT and the stromal adipogenic progenitor and T lymphocyte subsets in SAT. They also point out the stromal myofibrogenic progenitors as determinant in time-dependent adaptation to weight loss and metabolic improvement after BS.

The obese patients composing the SENADIP cohort exhibited a broad spectrum of metabolic impairment associated with obesity, with HOMA-IR index and HbA1c from normal to pathological ranges and 17% of the patients with type 2 diabetes. Similarly, a large range of responses to BS-promoted weight loss and IR improvement was observed. In agreement with what was already reported [21], this heterogeneity strongly suggests that adiposity itself is not the only determinant involved in the metabolic alterations associated with obesity and after weight loss. Increasing evidence suggests that the nature of the fat depot expansion also plays a major role. AT expands through the increase in adipocyte size and/or number [7]. Most of the studies describe a positive association between adipocyte hypertrophy and IR, but whether this association is distinct according to fat depot and specific to adipocyte subsets is controversial [22]. In the present study, the determination of the adipocyte repartition according to diameter shows that, despite higher percentages of adipocytes with 120 µm diameter (and more) in SAT compared to VAT, both HOMA-IR and HbA1c are associated with the percentages of adipocytes with 120 µm diameter (and more) in VAT. Therefore, the adipocyte hypertrophy in both SAT and VAT may be defined through a common threshold diameter but although higher percentages of adipocytes are hypertrophic in SAT, it is determinant in VAT only for metabolic flexibility. In agreement, diabetic obese patients of the SENADIP cohort were characterized by a shift in the repartition of the VAT adipocytes towards larger diameters as already reported by others [23]. Whether VAT adipocyte hypertrophy is the cause or consequence of IR remains to be clearly established, but hypertrophic adipocytes are considered as more lipolytic and resistant to the insulin-mediated anti-lipolysis effect, suggesting enhanced secretion of fatty acids in the portal circulation contributing to systemic IR [22]. In addition, adipocyte hypertrophy has been linked with inflammation. In agreement, recruited macrophages, which are positively associated with hyperglycemia, were also found in higher numbers in the VAT of diabetic obese patients, as well as B lymphocytes. Both immune cell populations are key cell players in AT inflammation and have been associated with IR. However, additional studies will be required to fully characterize their phenotypes and underlying mechanisms in the modulation of adipocyte function and whole metabolism homeostasis. Interestingly, only the hypertrophic VAT adipocyte phenotype was also negatively associated with the extent of weight loss and decreased HOMA-IR and HbA1c at 1 year and 3 years post-surgery and characterized patients who performed less in response to BS in terms of weight loss and metabolic flexibility. Therefore, it is suggested that VAT adipocyte hypertrophy, independently on immune-inflammatory cells, is the major VAT phenotype associated with impaired metabolic flexibility under both positive and negative energy balance. Additional data are needed to fully characterize the molecular mechanisms underlying such association, but it should be considered that the proportion of VAT adipocytes with intermediate diameter (70 µm to 90/100 µm) shows opposite associations prior to and after surgery than the ones observed with VAT hypertrophic adipocytes. It is therefore tempting to speculate that VAT adipocytes with intermediate sizes are protective against metabolic alterations and predictive of positive BS outcomes. The existence of subsets of mature adipocytes with distinct enriched metabolic and/or endocrine-related pathways has been clearly demonstrated with recent single-nuclei approaches [24]. Whether such differences are associated with adipocyte size remains to be established. Additional approaches allowing adipocyte sorting according to size [25] and/or spatial transcriptomics [26] will be necessary to clearly determine adipocyte size-dependent protective or deleterious phenotypes.

In case of sustained energy imbalance and maximal adipocyte hypertrophy, new adipocytes arise in fat depots. The consequent adipocyte hyperplasia increases the adipocyte storage capacity, notably in SAT that is predominant depot of the body [8], and therefore protects other organs from lipotoxicity [5]. Appearance of new adipocytes is due to the differentiation of local resident progenitor cells [26] towards adipogenesis. As for the mature adipocytes, single-cell RNA sequencing data have demonstrated that the AT progenitor cells are constituted by several subsets. Our previous studies using flow cytometry approaches highlighted three progenitor subsets based on the expression of additional cell surface markers MSCA1 and CD271 in human AT: the MSCA1 + progenitors characterized as the adipogenic precursor, the MSCA1 − /CD271 + (− /CD271 +) as the myofibroblast precursor and the subset negative for both MSCA1 and CD271 (−/−) as the more immature progenitor [4,27]. As already reported [4], higher numbers of adipogenic progenitors are found in SAT while higher numbers of myofibrogenic subset are found in VAT of obese patients. Prior to surgery, the number of adipogenic progenitors in SAT correlated positively with HbA1c and HOMA-IR. In addition, diabetic obese patients of the SENADIP cohort were characterized by elevated number of adipogenic progenitors in SAT. Therefore, the accumulation of adipogenic progenitors, which is more marked in SAT compared to VAT, is a characteristic of the expanded SAT of obese patients with impaired metabolic flexibility. The mechanisms underlying the accumulation of SAT adipogenic progenitors remain to be investigated but it is tempting to speculate that it may be related to a defect in the progression towards the process of adipogenesis. Our previous studies highlighted the local microenvironment and more particularly the T lymphocyte-derived inflammatory cytokines, including IFN-gamma, as a potent inhibitor of adipogenesis [27,28]. Despite higher numbers of B, NK and T lymphocytes in VAT compared to SAT of obese patients, the numbers of T lymphocytes in SAT only are positively associated with both HOMA-IR and HbA1c and the SAT of diabetic obese patients contains more helper T lymphocytes than non-diabetic obese individuals. It is thus tempting to speculate that the increased number of T lymphocytes in SAT may contribute, via the production of anti-adipogenic cytokines, to the accumulation of adipogenic progenitors. Altogether, the present data demonstrate that the accumulation of adipogenic progenitors together with T lymphocytes is the hallmark of unhealthy expansion of SAT in obesity. Defect of SAT adipogenesis may greatly contribute to SAT dysfunction by hindering the renewal of dysfunctional adipocytes but also to whole body energy homeostasis by limiting adipocyte hyperplasia and in consequence the SAT buffering capacity towards lipotoxicity [5]. Interestingly, the unhealthy SAT phenotype prior to surgery is also negatively associated with the outcomes of BS, showing that impaired remodeling capacity of SAT itself is a strong determinant under negative energy balance. Weight loss is the result of the reduced size of adipocytes with few to no changes in adipocyte numbers [29]. Therefore, it was expected that adipocyte hyperplasia prior to weight loss would predispose to higher risks of weight regain. The analysis of the weight loss or regain trajectories in response to BS showed no impact of the adipocyte hypertrophy nor of the numbers of adipogenic progenitors. However, the lower number of myofibrogenic progenitor subsets is the unique cell characteristic in both SAT and VAT that is associated with the most favorable weight loss trajectory. Additional investigations will be required to fully understand such a link; however, as adipocytes change size with weight loss and weight gain, a remodeling of the extracellular matrix is expected to occur in order to avoid mechanical stress upon adipocytes [30].

In conclusion, the present study demonstrates that the type of expansion of SAT and VAT is a major determinant for the maintenance of metabolic flexibility in obesity, but also in the variability of response to BS in terms of WL and improvement of IR. Restricted adipocyte hyperplasia in SAT, due to defect in adipogenesis under the negative influence of lymphocytes, together with VAT adipocyte hypertrophy are key actors. Therefore, the inter-depot rather than intra-depot balance between hypertrophy and hyperplasia appears to be a major determinant of metabolic health. In addition, the long-term adaptation of both SAT and VAT to negative energy balance probably involves the intrinsic fibrotic potential. The main limitation of the present study is the use of a single approach to characterize SAT and VAT stromal cells. The findings have to be confirmed with complementary unbiased approaches such as single-cell RNA sequencing, and the underlying mechanisms have to be explored, but they open up new avenues for identifying specific cell targets to improve metabolic flexibility and to understand the variability in the effectiveness of BS and thus better identify risk patients.

## Figures and Tables

**Figure 1 cells-11-03540-f001:**
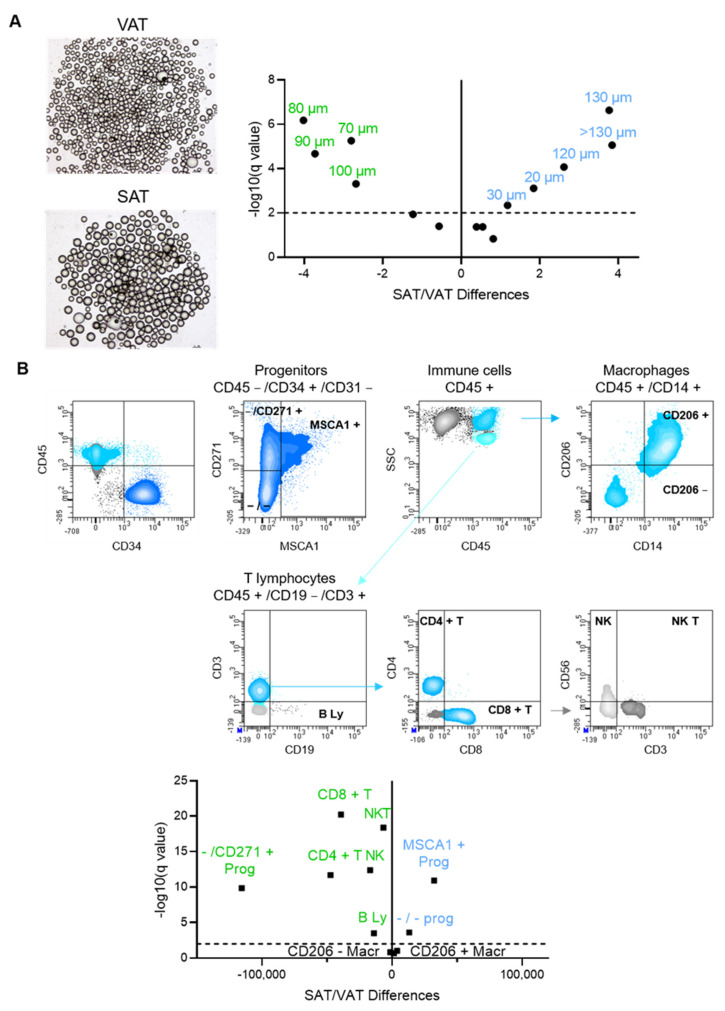
Differences in adipocyte size and stroma cell numbers between SAT and VAT. (**A**) Representative phase microscopy photography of VAT and SAT adipocytes are shown. Volcano plot of the differences in adipocyte diameter repartition (%) between SAT (blue) and VAT (green) according to −log10 (q value) for *n* = 101 paired tissues is shown. Numbers correspond to adipocyte diameter in µm. (**B**) Representative dot plots of the flow cytometry analyses are shown starting from viable cells gated on side scatter (SSC)/forward scatter (FSC) of the stroma vascular cells (not shown). Progenitor cells, defined as CD45 − /CD34 + /CD31 −, are gated from CD45/CD34 dot plot followed by CD31/CD34 dot plots (not shown). CD271/MSCA1 dot plot allows to further define the MSCA1 +, MSCA1 − /CD271 + (− /CD271 +) and MSCA1 −/CD271 − (−/−) progenitor subsets. Macrophages defined as CD45 + /CD14 +, are gated from SSC/CD45 +. Macrophage subsets are further defined based on CD206 expression. T lymphocytes defined as CD45 + /CD19 − /CD3 + are gated from CD45 + /low SCC, B lymphocytes identified as CD3 − /CD19 +. T lymphocytes are further divided into CD4 + helper T lymphocytes and CD8+ cytotoxic T lymphocytes. NK (CD3 − /CD56 +) and NKT (CD3 + /CD56 +) are gated from SSC/CD45 + cells. Volcano plot of the differences in stroma–vascular cell numbers in SAT (blue) and VAT (green) according to −log10 (q value) for *n* = 135 to 140 paired tissues is shown.

**Figure 3 cells-11-03540-f003:**
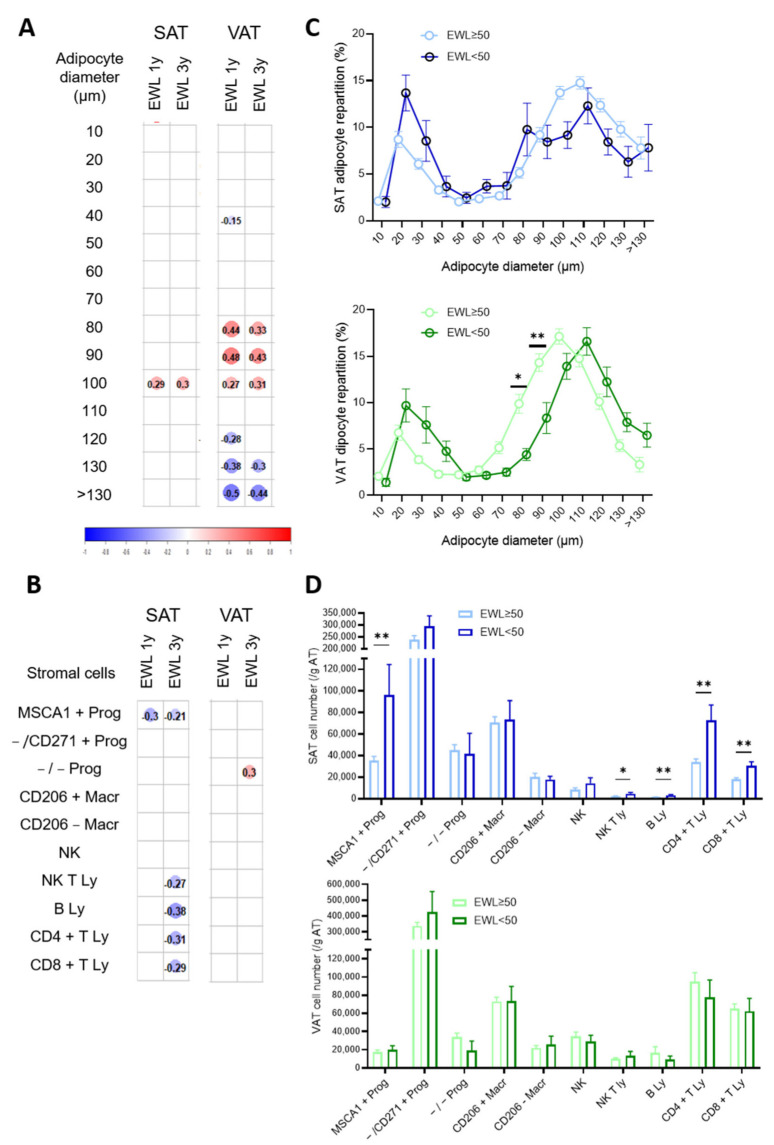
Cell phenotypes of expanded SAT and VAT associated with excess weight loss after BS. Correlation matrix between (**A**) adipocyte diameter repartition (%) in SAT and VAT or (**B**) stromal cell numbers in SAT and VAT prior bariatric surgery and excess weight loss (EWL) one year (1y) and three years (3y) post-surgery. Spearman correlations with *p* < 0.05 are shown with Spearman r value indicated in the color circle for *n* = 101 and *n* = 60 for adipocyte diameters at 1 year and 3 years, respectively, and *n* = 134 and *n* = 70 for stromal cells at 1 year and 3 years, respectively. (**C**) Adipocyte diameter repartition in SAT (blue) and VAT (green) of patients with EWL more or less than 50% three years post-surgery. Values are means ± SEM of *n* = 47 patients with EWL more than 50 and *n* = 9 with less than 50, two-way ANOVA followed by Sidak multiple comparisons test, *, *p* < 0.05, **, *p* < 0.01. (**D**) Stromal cell numbers determined by flow cytometry analyses in SAT (blue) and VAT (green) of patients with EWL more or less than 50% three years post-surgery. Values are means ± SEM of *n* = 57 − 58 patients with EWL more than 50 and *n* = 11 − 12 less than 50, multiple Mann–Whitney tests, *, *p* < 0.05.**, *p* < 0.01.

**Figure 4 cells-11-03540-f004:**
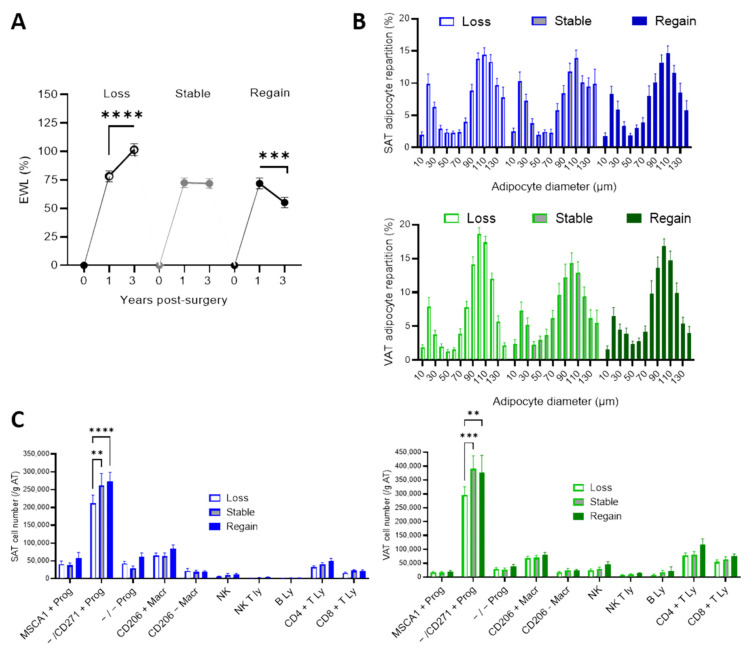
Trajectories of weight loss and cell phenotypes of expanded SAT and VAT. (**A**) Patients were grouped in tertiles (*n* = 26) according to the differences between EWL at three years and EWL at one year post-surgery with further weight loss (loss trajectory, stable weight loss (stable trajectory) and weight regain (regain trajectory); one way ANOVA followed by Tuckey multiple comparison test, ***, *p* < 0.001, ****, *p* < 0.0001. (**B**) Adipocyte diameter repartition in SAT (blue) and VAT (green) of patients according to weight loss trajectories. Values are means ± SEM of *n* = 18 to 20 patients. (**C**) Stromal cell numbers determined by flow cytometry analyses in SAT (blue) and VAT (green) of patients according to weight loss trajectories. Values are means ± SEM of *n* = 22 to 24 patients, two-way ANOVA followed by Sidak multiple comparisons test, **, *p* < 0.01, ***, *p* < 0.001, ****, *p *< 0.0001.

**Figure 5 cells-11-03540-f005:**
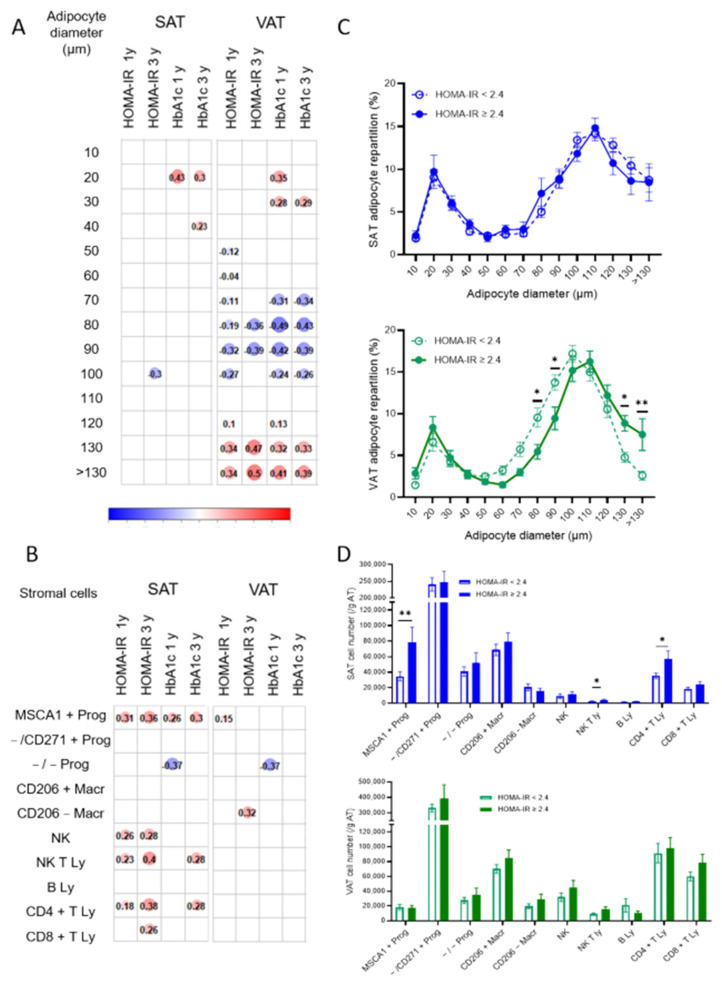
Cell phenotypes of expanded SAT and VAT associated with impaired metabolic flexibility after BS. Correlation matrix between (**A**) adipocyte diameter repartition (%) in SAT and VAT or (**B**) stromal cell numbers in SAT and VAT prior to bariatric surgery and HOMA-IR and HbA1c one year (1y) and three years (3y) post-surgery. Spearman correlations with *p* < 0.05 are shown with Spearman r value indicated in the color circle for *n* = 101 and *n* = 60 for adipocyte diameters at 1 year and 3 years, respectively, and *n* = 134 and *n* = 70 for stromal cells at 1 year and 3 years, respectively. (**C**) Adipocyte diameter repartition in SAT (blue) and VAT (green) of patients with HOMA-IR less than 2.4 (*n* = 32) or equal and more than 2.4 (*n* = 16) three years post-surgery *, *p* < 0.05, **, *p* < 0.01. (**D**) Stromal cell numbers determined by flow cytometry analyses in SAT (blue) and VAT (green) of patients with HOMA-IR less than 2.4 (*n* = 40) or equal and more than 2.4 (*n* = 17) three years post-surgery. Two-way ANOVA followed by Sidak multiple comparisons test, *, *p* < 0.05, **, *p* < 0.01.

**Figure 6 cells-11-03540-f006:**
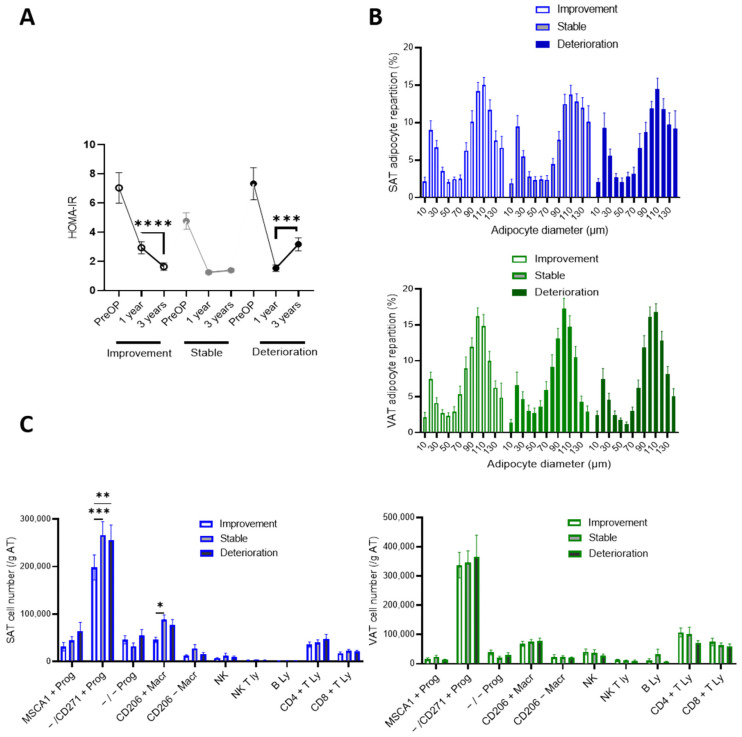
Trajectories of insulin resistance post-BS and cell phenotypes of expanded SAT and VAT. (**A**) Patients were grouped according to the tertiles (*n* = 26) of the differences between HOMA-IR at three years and HOMA-IR at one year post surgery with the first tertile qualified as improvement, intermediate tertile as stable and last tertile as deterioration. One-way ANOVA followed by Tuckey multiple comparisons test, ***, *p* < 0.001 ****, *p* < 0.0001. (**B**) Adipocyte diameter repartition in SAT (blue) and VAT (green) of patients with IR trajectories. Values are means ± SEM of *n* = 15 to 17 patients. (**C**) Stromal cell numbers determined by flow cytometry analyses in SAT (blue) and VAT (green) of patients prior to bariatric surgery with IR trajectories. Values are means ± SEM of *n* = 17 to 21 patients, two-way ANOVA followed by Sidak multiple comparisons test, two-way ANOVA followed by Sidak multiple comparisons test, *, *p* < 0.05, **, *p* < 0.01, ***, *p* < 0.001.

**Table 1 cells-11-03540-t001:** Pre- and post-surgery characteristics of the cohort.

Pre-Surgery (*n* = 161)	Mean
Age (years)	41.8
Weight (kg)	121.1
BMI (kg/m^2^)	44.2
HOMA-IR	5.6
HbA1c (%)	6
**1 Year Post-surgery (*n* = 153)**	
Total weight loss (%)	31
Excess weight loss (%)	74.3
HOMA-IR	1.9
HbA1c (%)	5.3
**3 Years Post-surgery (*n* = 78)**	
Total weight loss (%)	31.3
Excess weight loss (%)	76.5
HOMA-IR	2.1
HbA1c (%)	5.4

TWL (%) = weight loss in % of body weight, excess weight loss EWL = (initial weight – actual weight)/(initial weight – 25* height^2^); weight in kg and height in meters.

**Table 2 cells-11-03540-t002:** Cell surface markers of AT stromal cell subtypes.

Progenitors	CD45 − /CD34 + /CD31 −
Immature	MSCA1 − /CD271 −
Adipogenic	MSCA1 +
Myofibrogenic	MSCA1 − /CD271 +
**Immune** **cells**	CD45 +
NK cells	CD3 − /CD56 +
NKT cells	CD3 + /CD56 +
B lymphocytes	CD19 + /CD3 −
Helper T lymphocytes	CD19 − /CD3 + /CD4 +
Cytotoxic T lymphocytes	CD19 − /CD3 + /CD8 +
Resident macrophages	CD14 + /CD206 +
Recruited macrophages	CD14 + /CD206 −

**Table 3 cells-11-03540-t003:** Characteristics of patients with or without preoperative type 2 diabetes.

Characteristic	Without Diabetes (*n* = 119)		With Diabetes (*n* = 42)		*p* Value
	Mean	SD	Mean	SD	
Age (years)	39.3	10.2	48.9	10.1	<0.0001
BMI (kg/m^2^)	44.3	5.8	44	5.4	0.73
HOMA-IR	4.5	2.3	8.5	5.2	<0.0001
HbA1c (%)	5.6	0.4	7.1	1.2	<0.0001

Diabetes is defined as HbA1c equal or more than 6.5 and/or antidiabetic medication (biguanides 18; Sulfonylurea 7; 5; insulin 5, GLP1 R agonist 4, DPP-4 inhibitors 3). Statistical analyses were performed using multiple *t*-test.

**Table 4 cells-11-03540-t004:** Characteristics of patients with (or without) insufficient excess weight loss at 3 years post-surgery.

Characteristic	3 y EWL Success (*n* = 65)		3 y EWL Failure (*n* = 13)		*p* Value
	Mean	SD	Mean	SD	
3 y EWL (%)	85	25.4	35.2	12.1	< 0.0001
Age prior BS (years)	42	10	47	12	0.16
BMI prior BS (Kg/m^2^)	43.6	5.6	46.1	5.7	0.14
HOMA-IR prior BS	6	4.3	7.8	5	0.17
HbA1c prior Bs (%)	5.9	0.9	6.7	1.4	0.0064

EWL (Excess weight loss) success is defined as EWL equal or more than 50% and failure with EWL less than 50%. Statistical analyses were performed using multiple *t*-tests.

**Table 5 cells-11-03540-t005:** Characteristics of patients with (or without) IR improvement at 3 years post-surgery.

Characteristic	3 y IR Improvement (*n* = 49)		3 y Persistent		*p* Value
			IR (*n* = 19)		
	Mean	SD	Mean	SD	
3 y HOMA-IR	1.3	0.5	4	1.8	<0.0001
Age prior BS (years)	42.6	10	44.2	12.4	0.65
BMI prior BS (kg/m^2^)	44.1	5.8	44.9	5.9	0.63
HOMA-IR prior BS	4.7	2.3	10.4	5.8	<0.0001
HbA1c prior Bs (%)	5.8	0.6	6.6	1.2	0.0009

## Data Availability

Not applicable.

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
