# Peer review of "Subcutaneous Stromal Cells and Visceral Adipocyte Size Are Determinants of Metabolic Flexibility in Obesity and in Response to Weight Loss Surgery"

_cells, 2022, doi:10.3390/cells11223540_

Round 1

Reviewer 1 Report

The paper by Séverine Ledoux titled “Subcutaneous stromal cells and visceral adipocyte size are determinants of metabolic flexibility in obesity and in response to weight loss surgery” has been derived from a very well designed studied involving an extremely informative cohort of subjects. The use of the SENADIP cohort including consecutive patients with severe obesity who underwent Roux-en-Y gastric or bypass or sleeve gastrectomy in the same hospital, between April 2012 and October 2015, is indeed the main strength of the study.

The manuscript provides a lots of comparative data regarding adipocytes heterogeneity (and also about the SVF) and improvement in metabolic flexibility after the surgical procedure in a 3 years follow up.

The amount of data discussed is large and, in some part of the paper, not easy to follow and should be probably drained to allow readers to reach easily the take home message. Beyond adipocytes size, the markers used to characterize the different cell types are many and, if the readers are not completely involved in researches that use such a type of approach as experimental methods, they could get a bit lost.

The main limitation of the study is that it uses basically just a single technique (i.e. flow cytometry) to reach most of the conclusions drawn by the Authors. 

Also due to some limitation/difficulties reported in literature in the use of this technique on mature adipocytes (Cossarizza et al. 2017 Eur Journal of Immunology; Hagberg et al. 2018 Cell Report), it would have been nice to have also other clues for the phenotypic characterization of adipocytes and cells in the SVF.

For the reasons expressed above, it is not easy, to make specific comments to the experimental results presented in the manuscript. 

Overall, the study is well performed and the main massage is interesting and provide further evidences to support the idea of specific subsets of adipocytes with different physiological role in the pathophysiology of conditions such as insulin resistance and diabetes. The follow up, after bariatric surgery, in the cellular composition of the different fat depots is completely new field and it open new perspectives in the understanding of the underlying mechanisms regulating fat metabolism in light of the variability in the cellular composition.

Some speculations are present in the discussion and most of the conclusion are based on the assumption of adipocytes size as main determinant of the phenotype. 

I am aware that it might go beyond the scope of the present investigation, but the use of an alternative techniques to validate the phenotypic characterization and link it to some of the recent evidence present in the scientific literature (i.e. the report of three fat cell types with specific localization and mRNA/protein markers presented in the reference 24 of the list) it would have added strength to the conclusions.

Author Response

Reviewer 1

We thank the reviewer for her/his support and helpful comments. The manuscript has been corrected and we hope that we have addressed the reviewer concerns. Please find bellow our point by point answer.

  1. I suggest the authors to transfer the definition of persistent IR from the paragraph "Results", (page 12, Line 318) to the paragraph Materials and Methods, adding also the calculation of HOMA-IR.

We agree with the suggestion and modified the text in the paragraph Materials and Methods (lines 92-93) to include the calculation of HOMA-IR and the definition of persistent IR.

  1. The number of patients enrolled before BS was 161, but in table 2 the sum of patients without or with diabetes is 160. Is this correct?

The reviewer is right. The correct total number is 161 and the number of patients without diabetes in new table 3 is 119 (line 198).

  1. Also in Table 4, 68 of 78 patients were grouped into: 3-year improvement IR or 3-year persistent IR? What happened to the other 10?

We apologize for the lack of clarity. The gap between the number of subjects in new table 5 is due to missing data since HOMA index at 3 years was available in only in 68 subjects, unfortunately. This is now clearly indicated in the text line 330.

  1. In the Result Paragraph (Page 6 line 181) I think it is better to put… .. from 70 um to 90 um instead of… .. from 50 um to 90 um. As you said above, correlations are considered relevant for reduced metabolic flexibility when present for both parameters.

We agree with the reviewer and corrected the text line 188.

Reviewer 2 Report

Reviewer Comment.

I read with interest the manuscript "Subcutaneous stromal cells and visceral adipocyte size are determinants of metabolic flexibility in obesity and in response to weight loss surgery" by Séverine Ledoux and colleagues.

The study aimed to explore in a large cohort of patients with severe obesity the repartition of adipocyte diameters and the stroma progenitor subsets/immune cells in subcutaneous and visceral adipose tissue (SAT, VAT) and the relationships between SAT and VAT cell phenotypes with insulin resistance (IR) and diabetes pre bariatric surgery and weight loss and metabolic outcomes 1 year and 3 year post-surgery. The Authors found that the type of expansion of SAT and VAT is a major determinant for the maintenance of metabolic flexibility in obesity but also in the variability of response to BS in term of WL and improvement of IR.

This is a very well done manuscript dealing with an interesting subject. The results are clearly reported and well supported by a large experimental design. The data and analyzes are presented appropriately and the conclusions are interesting and provide an advance in current knowledge. The English form is linear and fluent.

There are only few minor comments to be address:

1. I suggest the authors to transfer the definition of persistent IR from the paragraph "Results", (page 12, Line 318) to the paragraph Materials and Methods, adding also the calculation of HOMA-IR.

2. The number of patients enrolled before BS was 161, but in table 2 the sum of patients without or with diabetes is 160. Is this correct? Also in Table 4, 68 of 78 patients were grouped into: 3-year improvement IR or 3-year persistent IR? What happened to the other 10?

3. In the Result Paragraph (Page 6 line 181) I think it is better to put… .. from 70 um to 90 um instead of… .. from 50 um to 90 um. As you said above, correlations are considered relevant for reduced metabolic flexibility when present for both parameters.

Author Response

We thank the reviewer for her/his valuable critique and suggestions. We corrected the text to clarify the manuscript and we hope to have addressed all of the comments to the reviewers' satisfaction. Please find bellow our point by point answer.

The amount of data discussed is large and, in some part of the paper, not easy to follow and should be probably drained to allow readers to reach easily the take home message.

Beyond adipocytes size, the markers used to characterize the different cell types are many and, if the readers are not completely involved in researches that use such a type of approach as experimental methods, they could get a bit lost.

As the reviewer suggested, we add a table summarizing the correspondences between the cell surface markers and the stromal cell subsets in new table 2 (line 127). We hope that this new table will help the readers to better identify the different cell types. In addition, we took care of using generic cell names mentioned in new table 2 instead of the combination of cell surface markers in the main text in order to allow a more fluent reading.

The main limitation of the study is that it uses basically just a single technique (i.e. flow cytometry) to reach most of the conclusions drawn by the Authors. Also due to some limitation/difficulties reported in literature in the use of this technique on mature adipocytes (Cossarizza et al. 2017 Eur Journal of Immunology; Hagberg et al. 2018 Cell Report), it would have been nice to have also other clues for the phenotypic characterization of adipocytes and cells in the SVF.

We already validated by numerous publications the use of flow cytometry to identify and to quantify adipose stromal cells but we agree with the reviewer that the main limitation of the study is that the identification of the cell subsets relies on a single approach. We have now included this statement in the text line 485 and added as suggested by the reviewer that the findings have to be confirmed with complementary unbiaised approaches such as single cell RNA sequencing” lines 486-487. In addition, we add the reference suggested by the reviewer “Hagberg et al. 2018 Cell Report” in the line 427. To note, histological approaches also highlighted link between mean adipocyte area and weight loss after bariatric surgery (Lemoine AY, Ledoux S, et al. JCEM. 2012 May;97(5):E775-80. PMID: 22419723).

Some speculations are present in the discussion and most of the conclusion are based on the assumption of adipocytes size as main determinant of the phenotype. I am aware that it might go beyond the scope of the present investigation, but the use of an alternative techniques to validate the phenotypic characterization and link it to some of the recent evidence present in the scientific literature (i.e. the report of three fat cell types with specific localization and mRNA/protein markers presented in the reference 24 of the list) it would have added strength to the conclusions.

In agreement with the reviewer suggestion, we modified the text in the discussion part, line 420 to clearly indicate the speculative part concerning adipocyte size. Moreover, we add a discussion part on the additional approaches to be performed to state about adipocyte size-dependent protective or deleterious phenotypes (lines 424-428) and two additional references on flow cytometry (Hagberg et al) and spatial transcriptomics (Backdahl et al) have been included.